# Highly-efficient electrically-driven localized surface plasmon source enabled by resonant inelastic electron tunneling

Haoliang Qian [1,2,6], Shilong Li[1,6], Su-Wen Hsu [3,6], Ching-Fu Chen[1], Fanglin Tian[1], Andrea R. Tao[3,4] & Zhaowei Liu [1,4,5 ✉]

On-chip plasmonic circuitry offers a promising route to meet the ever-increasing requirement for device density and data bandwidth in information processing. As the key building block, electrically-driven nanoscale plasmonic sources such as nanoLEDs, nanolasers, and nano-junctions have attracted intense interest in recent years. Among them, surface plasmon (SP) sources based on inelastic electron tunneling (IET) have been demonstrated as an appealing candidate owing to the ultrafast quantum-mechanical tunneling response and great tunability. However, the major barrier to the demonstrated IET-based SP sources is their low SP excitation efficiency due to the fact that elastic tunneling of electrons is much more efficient than inelastic tunneling. Here, we remove this barrier by introducing resonant inelastic electron tunneling (RIET)—follow a recent theoretical proposal—at the visible/near-infrared (NIR) frequencies and demonstrate highly-efficient electrically-driven SP sources. In our system, RIET is supported by a TiN/Al$_2$O$_3$ metallic quantum well (MQW) heterostructure, while monocrystalline silver nanorods (AgNRs) were used for the SP generation (localized surface plasmons (LSPs)). In principle, this RIET approach can push the external quantum efficiency (EQE) close to unity, opening up a new era of SP sources for not only high-performance plasmonic circuitry, but also advanced optical sensing applications.

[1] Department of Electrical and Computer Engineering, University of California, San Diego, 9500 Gilman Dr, La Jolla, CA, USA. [2] Interdisciplinary Center for Quantum Information, State Key Laboratory of Modern Optical Instrumentation, ZJU-Hangzhou Global Scientific and Technological Innovation Center, Zhejiang University, Hangzhou, China. [3] Department of NanoEngineering, University of California, San Diego, 9500 Gilman Dr, La Jolla, CA, USA. [4] Materials Science and Engineering Program, University of California, San Diego, 9500 Gilman Dr, La Jolla, CA, USA. [5] Center for Memory and Recording Research, University of California, San Diego, 9500 Gilman Dr, La Jolla, CA, USA. [6]These authors contributed equally: Haoliang Qian, Shilong Li, Su-Wen Hsu. ✉email: zhaowei@ucsd.edu

Over the last few decades, plasmonics has brought us new opportunities to control light at length scales previously thought impossible[1–5], thus leading to many inspiring applications such as super resolution imaging that breaks the diffraction limit[6,7], new kind of optical sensors with enhanced performance[8,9], and electromagnetic metamaterials by which one can manipulate light-matter interaction in an unforeseen degree of freedom[10,11]. With the development of nano-fabrication techniques, on-chip electrically-driven plasmonic circuitries have gained great attention[12–15] in the post-Moore era for they hold the promise to combine both the small device footprint (<10-nm feature size) of electronic circuitry and the large information capacity (>100-THz bandwidth) of a photonic network. A variety of plasmonic building blocks[12–22] have been demonstrated ranging from sources, waveguides, routers, modulators, to detectors. So far, the widely used electrical source[12,13,17–19] of surface plasmons (SPs) relies on a two-step process, namely the generation of photons by electrically-triggered spontaneous emission and the subsequent SP excitation via near-field coupling of the generated photons. While the Purcell effect accelerates this spontaneous-emission process, the modulation speed of these plasmonic sources is still limited (>1 ps)[14]. On the other side, the modulation rate of plasmonic nanolasers—light sources that rely on the stimulated emission process—is also within the sub-THz range[23]. In addition, these plasmonic nanolasers' emissions are almost fixed at one specific frequency by their design. Recently, direct electrical excitation of SPs via inelastic electron tunneling (IET) in metal-insulator-metal (MIM) junctions has reemerged as a promising ultrafast source to drive the integrated plasmonic circuitries[14,15,24,25]. Since this quantum-mechanical tunnel event is governed by Heisenberg's uncertainty principle, IET-based plasmonic sources could have a temporal response as fast as few fs at the visible/near-infrared (NIR) frequencies[26–29], limited only by the RC time constant of the sources. Thus, an IET-enabled electrically-driven SP source provides the most appealing strategy for addressing the requirement in bandwidth promised by plasmonic circuitries. However, the major issue of the IET-based source is its low SP excitation efficiency due to the dominant elastic tunneling events of electrons.

So far, the electro-plasmon transduction efficiency—described by external quantum efficiency (EQE)—has been improved by tailoring the local density of optical states (LDOS) of an IET source[14,30,31]. However, optical engineering of the LDOS of IET sources has hit its limit set by quantum-mechanical effects such as electron tunneling and nonlocal screening in plasmonic nanostructures[31,32]. Nevertheless, further improvement of the EQE is possible via engineering the electric portion of the IET source. For example, a recent theoretical calculation[33] showed that the efficiency of SP excitation by IET could be up to 100% in a resonant tunneling structure for electrons, such as a metal-insulator-metal-insulator-metal (MIMIM) multilayer where the middle, ultrathin (few nanometers) conducting layer forms a quantum well (QW) due to confinement from the adjacent insulating barriers. The basic idea is to impede the elastic tunneling of electrons across the MIMIM multilayer by the two insulator layers, and, at the same time, to facilitate their inelastic tunneling process via resonant electron states of the middle QW structure; the similar idea was widely used on the resonance effects in photon assisted electron transport studies for semiconductors[34,35]. However, the requirement of such a few-nanometer-thick metal/dielectric heterostructure—aiming at the technologically more important visible/NIR spectral range for on-chip integrated photonics applications—faces large challenges in fabricating such structures.

In this article, we demonstrate an efficient on-chip electrically-driven resonant inelastic electron tunneling (RIET)-based SP source at the visible/NIR frequencies in metallic quantum well (MQW)-based tunnel junctions. Without loss of generality, we focus on the localized surface plasmon (LSP) source in the current RIET demonstration; the same RIET mechanism is readily applicable for non-LSP sources such as the surface plasmon polariton (SPP) source; moreover, with proper nanoantenna design[36], one might also achieve high-performance RIET-based photon sources.

## Results

**Visible/NIR frequency RIET enabled by MQWs.** As schematically shown in Fig. 1a, the RIET SP source demonstrated in this work consists of an MQW junction biased through the upper conducting ITO and the lower metallic TiN, with silver nanorods (AgNRs) on top of the MQW. The transmission electron microscopy (TEM) cross-section of the MQW is shown in Fig. 1c, d, where an ultrathin metal film (~1.4-nm TiN) with atomically flat interfaces in-between dielectrics (~10 nm for each $Al_2O_3$) over a large area is clearly seen (see the fabrication method in Supplementary Note 1). Such high-quality dielectric/ultrathin-metal/dielectric sandwich heterostructures are crucial in keeping the quantum size effect; as a result, the conduction band of this ultrathin TiN is split into resonant electron subbands[37–39]. The calculated electronic subband diagram of this MQW is given in Supplementary Note 2, which shows that there are seven discrete states provided by the MQW owing to its high QW barrier (~8 eV), allowing one not only to realize RIET at visible/NIR frequencies, but also to tune it by a large range of external voltages. When the resonant electron state $|5\rangle$ of the middle MQW is aligned with the Fermi level of the TiN positive electrode by increasing the bias voltage (Fig. 1b), RIET processes start—i.e., electrons in the ITO negative electrode tunnel through this MQW junction inelastically by coupling to a plasmonic mode of energy $h\nu$ supported by the AgNRs on top of the junction (Fig. 1a, b), where $h$ is the Planck constant and $\nu$ is the frequency. The resonant electron states provided by the high-quality MQW tunnel junction enable RIET in visible/NIR frequencies that can be used for highly efficient electrically-driven SP sources.

**Device structure.** Figure 2a presents the structural cross-section of RIET SP sources. It is divided into four regions in order to carry out both electrical pumping and optical far-field measurement on a single device mesa. Electrons are injected from the external circuitry into an Au layer at region *iii*; then, they are transported into the top ITO layer at region *ii*, and are accumulated on top of an MQW junction at region *i*. As shown in Fig. 1b, these electrons can tunnel inelastically through this junction by coupling to an AgNR LSP mode, and then go back to the external circuitry across the underlying TiN layer. Scanning electron microscope (SEM) images in Fig. 2b, c show the top view of the RIET devices; the different regions of one device mesa can be distinguished, while the different device mesas are separated by region *iv*—manifesting themselves as an integrative, scalable SP device. A small fraction of the generated LSPs will be scattered off—i.e. converted to photons—from the AgNR to the far field through the top transparent ITO layer in region *i*, enabling an optical far-field measurement on the SP source.

**Measurement and simulation of optical responses.** Monocrystalline AgNRs were used to support LSPs for the RIET source. Figure 2d is an exemplary SEM image showing an array of AgNRs homogenously distributed over region *i* before ITO deposition. These AgNRs, with a diameter of 40 ± 5 nm and a length of 130 ± 10 nm (Fig. 2e), were organized into this loosely-packed monolayer at an air-water interface and then transferred on top of the MQW junction

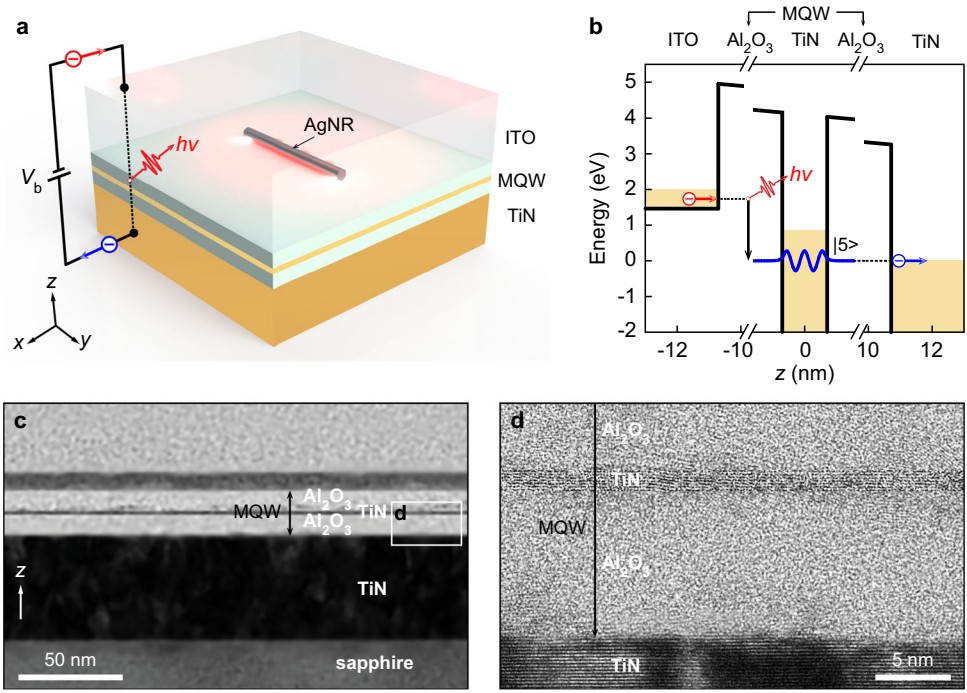

**Fig. 1 RIETs enabled by an MQW for highly-efficient electrically-driven SP sources. a** Schematic drawing of the electrically-driven SP source (not to scale). An insulating MQW junction is biased via the upper conducting ITO and the lower metallic TiN. Electrons may inelastically tunnel through this MQW junction by coupling to a plasmonic mode of energy $h\nu$ supported by AgNRs on top of the junction. **b** RIET via the MQW junction. In this MQW, the resonant state $|5\rangle$ is responsible for the RIET process that emits SPs with the energy $h\nu$ in the visible/NIR spectral range. **c** Transmission electron microscopy (TEM) cross-section of an MQW heterostructure. The layers on top of the MQW are the protective layers used only for TEM cross-section preparation during the focused ion beam (FIB) cutting process. **d** High-resolution TEM (HRTEM) image showing an enlarged view of the MQW.

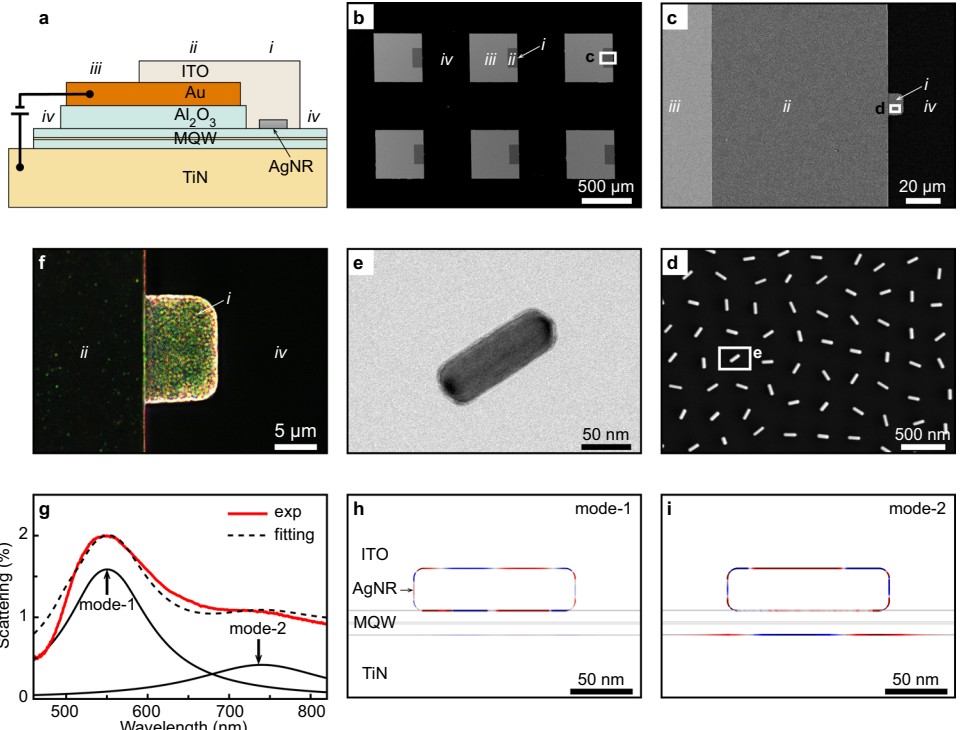

**Fig. 2 Structural characterization and optical property. a** Schematic drawing of the structural cross-section of RIET SP sources (not to scale). For the sake of illustration clarity, four regions (indicated by Roman numerals *i–iv*) are marked for each of the device mesas ($7 \times 7$) of a sample ($10 \times 10$ mm$^2$). **b–d** SEM images showing device mesas (**b**), regions of one mesa (**c**), and AgNRs in region *i* before ITO deposition (**d**). **e** TEM image of a single AgNR. **f,g** Dark-field optical image (**f**) and scattering spectrum (**g**) of region *i* with the ITO. Two plasmonic resonance modes are superimposed in the scattering spectrum as solid black curves. **h,i** Simulated charge distributions of the two LSP modes shown in **g**. Red and blue colors represent positive and negative charges, respectively.

(Supplementary Note 1) via Langmuir-Blodgett deposition. A dark-field optical image of region $i$ is shown in Fig. 2f where diffraction-limited spots due to scattering by the AgNRs are clearly visible, and their greenish color corresponds to the intense resonance peak (centered at the wavelength of ~550 nm) of the scattering spectrum taken from this region (Fig. 2g). Finite element method (FEM) simulations (Supplementary Note 3) were conducted, and the simulated scattering spectrum under a plane-wave excitation agrees well with the experimental one (Fig. 2g). The charge distributions of two LSP modes (see all the SP modes in Supplementary Note 3) are shown in Fig. 2h, i. As can be seen, the mode centered at the shorter wavelength of ~550 nm results mainly from the longitudinal octu-polar resonance—i.e. the $N = 3$ LSP mode (Fig. 2h), where the positive integer $N$ is the longitudinal mode index[40]; the longer-wavelength (~740 nm) mode arises from both the longitudinal quadrupolar ($N = 2$) resonance and the gap plasmon resonance between the AgNRs and the underneath TiN metallic layer (Fig. 2i). As will be seen below, these two LSP modes determine not only the scattering spectrum of the RIET-based SP source but also its emission spectrum.

**Measurement and comparison of electro-optical responses.** To gain a better understanding on the electro-optical response of the RIET device, we fabricated control samples that were not covered with AgNRs. Typical current–voltage ($I$–$V$) curves of a RIET device and its control sample are shown in Fig. 3a. A current peak (peak-1) centered at ~1.3 V was observed for both the two devices. It gives rise to a negative differential resistance region,

which is the signature of a resonant tunneling diode[41], indicating that the fabricated MQWs indeed provide resonant electron states. However, this resonant elastic electron tunneling (REET) process (Fig. 3d) is unfavorable because it severely reduces the SP generation efficiency. Nevertheless, there is no more REET pro-cess at voltages above ~1.6 V, as can be seen from the $I$–$V$ curve of the control sample (Fig. 3a). It is worth noting that the AgNRs were encapsulated by a polyethylene glycol (PEG) ligand shell; the thickness of the ligand shell is about 1–2 nm (Fig. 2e) and thus it electrically insulates the AgNRs from the external circuitry (see the equivalent circuit in Supplementary Note 4); this isolation property of ligand shell has been utilized to demonstrate an IET-based light source[36] where the ligand shell of silver nanocubes was used as the electron tunneling barrier.

Apart from the low-voltage REET peak, two more current peaks (peak-2 and peak-3) at high voltage were only observed for the RIET device with AgNRs. We attribute these two high-voltage peaks to a large tunneling current accompanied by an intense RIET process. This is evidenced by inspecting the voltage dependence of the corresponding LSP emission power, as shown in Fig. 3b. In this high-voltage range, electrically-driven LSPs were generated, propagated, and scattered at the end of the AgNRs, and then converted to far-field photons; a typical image of photon emission at $V_b = 2.2$ V is shown in Fig. 3e, where diffraction-limited spots due to scattering by the AgNRs are clearly visible. These far-field photons were collected to estimate the EQE of the RIET SP source (Supplementary Note 4 and Note 5). Figure 3c shows the voltage dependence of the EQE of the RIET SP source, biased at a voltage above the REET current

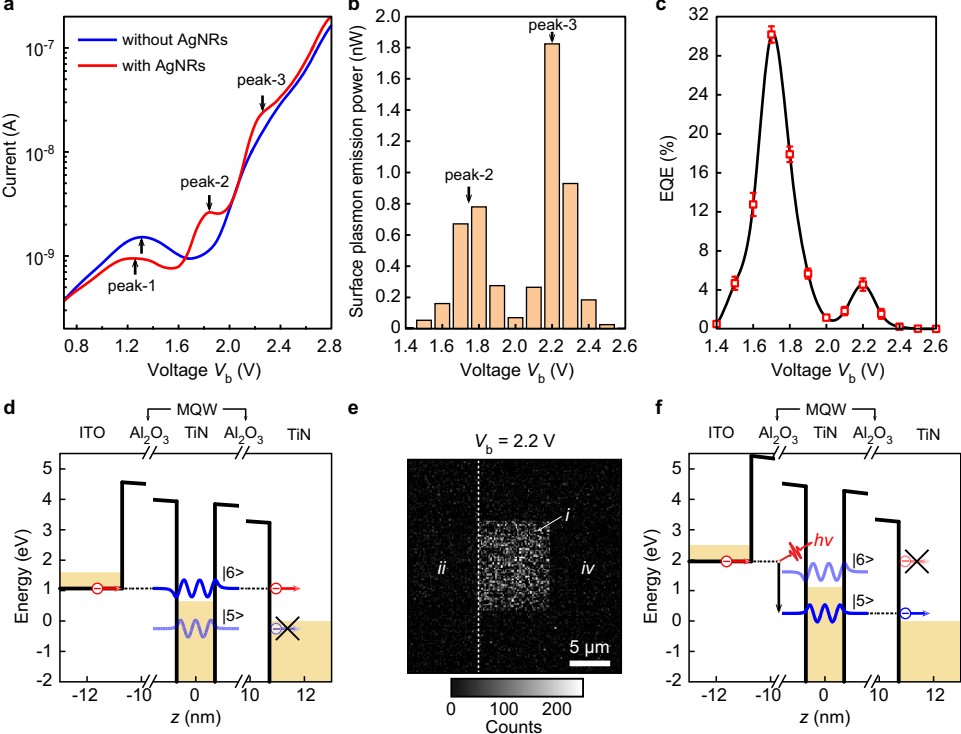

**Fig. 3 Electro-optical characterization. a** $I$–$V$ curves of a RIET device (with AgNRs) and its control sample (without AgNRs). **b** Voltage dependence of LSP emission power. **c** Voltage dependence of the EQE. The error bars were obtained from multiple experimental measurements. **d** REET process at peak-1 in **a**. The REET current reaches its peak when the conduction band of ITO is aligned with the resonant state $|6\rangle$ of the MQW. The resonant state $|5\rangle$ is now below the Fermi level of the right electrode, so that the corresponding RIET process is forbidden. **e** Image of far-field photon emission at $V_b = 2.2$ V. **f** RIET process at peak-3 in **a**. The REET process of the electron via the resonant state $|6\rangle$ is suppressed, while its RIET process by the resonant state $|5\rangle$ is active to excite the LSP. These two factors together lead to the high EQE obtained in **c**. Note that the RIET via the resonant state $|5\rangle$ is also possible; however, it occurs at the mid-infrared wavelength which is out of our current plasmonic design and experimental window, so the effect of this RIET process can be ignored.

peak (i.e., peak-1 in Fig. 3a). Two EQE peaks were observed in this high-voltage range. Their position (peak voltage) is determined by the optical and electrical processes involved in RIET. It is worth noting that one can also find the origin of the two current peaks by analyzing the second derivative of the I–V curve, as discussed in Supplementary Note 6, where similar discussions on the control sample without MQWs and the RIET device with reversed bias are given as well.

**Analysis on the EQE.** The peak EQE reaches up to 30%, which is, to the best of our knowledge, the highest value demonstrated thus far for an on-chip electrically-driven SP generation. To have a simple physical picture for this high EQE, we can express the EQE as the product of an internal quantum efficiency (IQE) for the electro-optical transduction and a near-unity excitation probability of SPs (Supplementary Note 5), where the IQE is the ratio $\Gamma_{inel}/(\Gamma_{inel} + \Gamma_{el})$ with the inelastic (elastic) transition rate $\Gamma_{inel}$ ($\Gamma_{el}$) of electrons. Therefore, the EQE is solely determined by the corresponding IQE, and, in order to gain a high EQE, one has to increase $\Gamma_{inel}$ while decrease $\Gamma_{el}$. It is worth noting that this requirement cannot be fulfilled with a single-barrier tunnel junction as used in previous demonstrations[14,36,42,43]. However, it can be done with the MQW junction: As shown in Fig. 3f, the decrease of $\Gamma_{el}$ is realized, on the one hand, by the use of two thick barriers (10-nm $Al_2O_3$ on each side) and, on the other hand, by suppression of the REET process at high voltages; at the same voltage, however, the increase of $\Gamma_{inel}$ is achieved via the RIET process. Ideally, the EQE of a RIET SP source may approach 100%, as theoretically proposed[33]. However, in reality, the dark current of the RIET device is not negligible compared to the tunneling current companied by the RIET process (Fig. 3a), which limits the EQE to ~30% in the current devices. There are several possible reasons responsible for this non-negligible dark current. First, it might be caused by the leakage of the current via defects in the nonperfect $Al_2O_3$ insulating layers. Secondly, the interface roughness that relaxes conservation of in-plane momentum and thus increases elastic tunneling[44] could also cause the dark current. The second reason would be the residual REET current due to the broadening of the REET process resulted from the ultrashort resonant state lifetime. One more reason could be due to light emissions into other non-SP ultrafast decay channels, such as intersubband transition (ISBT) modes of the MQW[37], and nonradiative inelastic tunneling with phonon emission—similar to nonradiative relaxation in quantum cascade lasers[45]. It is also worth to note that the method of a far-field measurement combining with the simulation of the far-field radiation efficiency used in this work for the RIET-based SP source has widely been utilized to determine the EQE of IET-based photon[36,42,43,46] and SP[14,47,48] sources.

**Theoretical model for RIET SP sources.** Figure 4a shows the SP emission spectra of a RIET device at varied voltages, which were obtained from the corresponding far-field emission spectra (see details in Supplementary Note 4). There is a cutoff frequency $\nu_{max}$ at a given voltage. It is a signature of IET-based sources[49], and can be described by the quantum relation $h\nu_{max} = eV_b$, where e is the electron charge. The SP radiation power spectrum can be expressed as[26,30,31]

$$P_{SP}(\nu, V_b) = h\nu \times \gamma_{inel}^0(\nu, V_b) \times \frac{\rho_{opt}}{\rho_0} \times \eta_{SP}(\nu) \quad (1)$$

where $\gamma_{inel}^0$ is the spectral inelastic transition rate in vacuum, $\rho_0$ is the vacuum LDOS, $\rho_{opt}$ is the device LDOS, and $\eta_{SP}$ is the SP radiation efficiency. The spontaneous emission model developed for IET-based sources was used to calculate $\gamma_{inel}^0$ (Supplementary

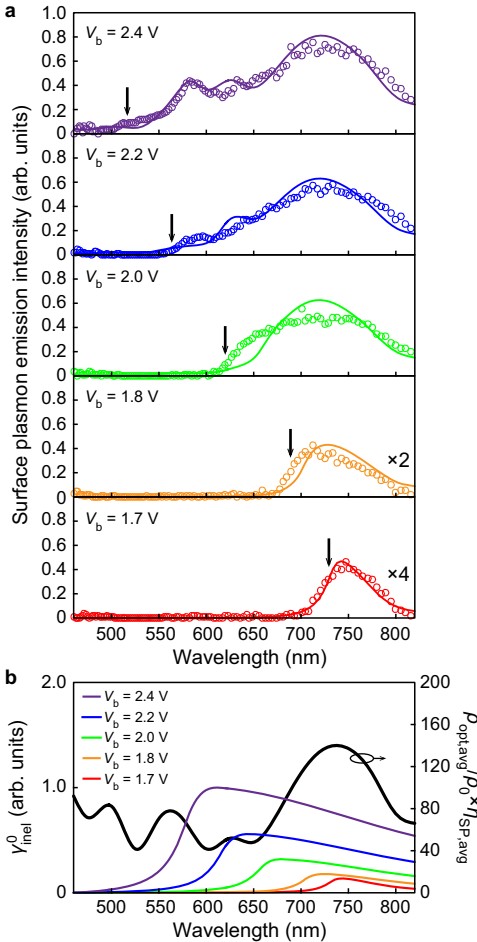

**Fig. 4 Tunable SP emission spectrum. a** Voltage dependence of the SP emission spectrum (colored dots) for a RIET device. For clarity, data at the two low voltages 1.7 V and 1.8 V are amplified by a factor of 2 and 4, respectively. Solid colored lines are the respective SP radiation power spectrum calculated based on the model described in the text, where the same scaling factors were used to match the experimental results. Black vertical arrows indicate the respective cutoff frequency $\nu_{max}$ with the quantum relation $h\nu_{max} = eV_b$. Note that the small portion of emission with $\nu > eV_b/h$ could be attributed to multielectron processes or hot-electron light emission[50,51]. **b** Calculated vacuum source efficiency $\gamma_{inel}^0$ (color lines) and SP radiation enhancement $\rho_{opt,avg}/\rho_0 \times \eta_{SP,avg}$ (black line) at various voltages. Here, $\rho_{opt,avg}$ and $\eta_{SP,avg}$ were obtained under a dipole-source excitation with different polarization states and spatial positions, see Supplementary Note 3 for details.

Note 2), FEM simulations were applied to obtain the LDOS enhancement $\rho_{opt}/\rho_0$ (Supplementary Note 3), while $\eta_{SP}$ is determined by the ratio of the SP excitation power $p_{sp}$ to the total power dissipation $p_{tot}$ as

$$\eta_{sp} = \frac{p_{sp}}{p_{tot}} = \frac{p_{sp}}{p_{sp} + p_{loss} + p_r} \quad (2)$$

where the total dissipated power consists of not only the SP excitation part $p_{sp}$, but also the absorption loss $p_{loss}$ by the device materials and the far-field radiative part $p_r$ (Supplementary Note 3). It is worth noting that, since we are aiming for a SP source, the power dissipated is designed to convert mainly to near-field SPs, but not to far-field photons[36,46]; thus, the far-field radiation efficiency $\eta_r = p_r/p_{tot}$ is low—on the order of $10^{-3}$ (Supplementary Note 3). The calculated SP radiation power

spectra $P_{SP}(\nu, V_b)$ are superimposed in Fig. 4a. This plot shows how this theoretical model sufficiently matches our experimental observations. As shown in Eq. 1, these spectra result from both the electrical properties (i.e. $\gamma_{inel}^0(\nu, V_b)$) and optical response (i.e. $\rho_{opt}/\rho_0 \times \eta_{SP}(\nu)$) of the RIET source, and their wavelength dependencies are shown in Fig. 4b. As can be seen, the plasmonic resonances of AgNRs determine the spectral peaks, which come out only when the applied bias exceeds the corresponding plasmon mode energy, and are modulated by the wavelength-dependent RIET process.

## Discussion

Although the focus of this work is on the EQE, the total power emitted is also an important figure of merit for a practical source and thus given here: The emission power of the RIET plasmon source is on the order of 1 nW (see Supplementary Note 4 for the discussion on emission powers). It is also worth noting that the large area of homogeneously distributed AgNRs were used for the purpose of demonstrating the high EQE LSP source enabled by RIETs in MQWs; for the further applications in plasmonic circuitries, the RIET source consists of essentially a single AgNR (see Supplementary Note 7 for the discussion on plasmonic circuitry integration).

In conclusion, we have demonstrated highly-efficient RIET-based SP sources with an MQW junction. The large well-depth of the MQW provides a plenty of resonant electron states with transition energies covering the entire visible/NIR frequency range, allowing on-chip plasmonic circuitries for optical communications and information processing in the desired operating window. The working frequency of the RIET source is fully determined by the applied external voltages, resonant tunneling configurations, and the designed LDOS, exhibiting its broadband tunability. All these experimental observations are well explained and modeled. Future optimization includes selecting less lossy plasmonic materials, improving the barrier material quality, and reducing the fabrication related defects, which will further boost the EQE. Various application-oriented RIET-based LSP, SPP and photon sources could be realized by designing proper optical micro-/nanostructures. The demonstrated high-efficiency, tunable SP source would thus enable on-chip active plasmonic sensors, and eventually push nanoscale electronic-plasmonic systems toward much faster yet efficient computing frameworks.

## Methods

**Numerical simulation**. Numerical finite-element simulations were performed with the COMSOL Multiphysics. The numerical calculation of electron resonant states in metallic quantum wells is provided in Supplementary Note 2 in details. The numerical simulations of SP radiation efficiency and optical far-field radiation are provided in Supplementary Note 3 in details.

**Sample preparation**. The fabrication of resonant tunnel junction is provided in Supplementary Note 1 in details.

**Sample characterization**. The electrical and optical measurement setups for the resonant inelastic tunnel devices are provided in Supplementary Note 4 in details.

## Data availability

The authors declare that the data supporting the findings of this study are available from corresponding authors upon reasonable request.

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

## Author contributions

H.Q. and Z.L. conceived the idea. S.L. and H.Q. performed the theoretical calculation and numerical simulation. H.Q., C.C. and F.T. performed the metallic quantum well growth. S.H. prepared and characterized the Ag nanocrystals. H.Q. and S.H. performed the device fabrication. H.Q. performed the sample characterization and light generation experiments. S.L., H.Q., A.T. and Z.L. wrote the manuscript. All authors analyzed the data and revised the manuscript. Z.L. supervised the research.

## Competing interests

The authors declare no competing interests.
