## [Peer Review File · Nature Communications]

REVIEWER COMMENTS

Reviewer #1 (Remarks to the Author):

The paper has been revised and the main claim partially changed. I recommend publication after a few mandatory changes.

1) There is still a confusion in the paper already pointed out by referees 1 and 2 in the first round. The motivation put forward is plasmonic circuitry which requires surface plasmons polaritons (SPPs) whereas the EQE reported here only deals with localized surface plasmons (LSPs).

The process of transforming a LSP into a SPP comes with a coupling cost. Using a nanowire supporting SPPs is not shown according to the supplementary because of low signal. Thus, this is not a detail. It is necessary to point out that the EQE reported here deals with LSP so that SP must be replaced everywhere in the text by LSP when it refers to the nano rods.

2) A second necessary modification is to discuss the uncertainty on the EQE estimation. There are no bar errors in the discussion in the supplementary.

Reviewer #2 (Remarks to the Author):

After studying the responses of the authors to my criticisms, I could see that the changes introduced into the revised version are minor: there is still the efficiency (EQE) claim of 30%, which is now found not in the abstract but in the main text and Fig. 3c, still supported only by simulations indicating a very low far-field radiation efficiency. As was emphasized in my review, there are no propagating surface plasmons being excited, only localized ones, but the authors still write on p. 9 (and implicitly assume elsewhere) that "electrically-driven SPs were generated, propagated, and scattered at the end of the AgNRs". The prospects of integration with a plasmonic circuitry are still only prospects described in a very short and highly speculative Supplementary Section S7, which is based on the prospect of realizing an RIET source based on a single AgNR. The latter would indeed be a very important development, but its realization is still missing. Finally, the bandwidth measurements in any form are still missing.

Overall, in my opinion, the level of the obtained results is not sufficient for this work to be recommended for the publication in Nature Communications.

Reviewer #3 (Remarks to the Author):

The review on the paper by H. Qian et al "Highly-efficient electrically-driven surface plasmon source enabled by resonant inelastic electron tunneling"

As I stated previously, the work of H. Qian et al reports on a very interesting experimental observation of the plasmon excitation in the tunneling metal-dielectric-metal (quantum well)-dielectric-metal device, where the elastic tunneling is suppressed and only the inelastic tunneling is possible. In an

ideal system without losses each tunneling event can lead to the electromagnetic excitation, and, through the careful design of the electromagnetic density of states, to the plasmon excitation of the nanoantenna.

While the resonant structures alike are known in semiconductor physics (obviously at much lower electromagnetic radiation frequencies), present application for metal plasmonics is a primer. Already at the very beginning, I strongly supported publication of this work for the beauty of the physics and for the thoughtful design of the structure. Now, when the authors addressed comments of the referees twice, I think that the paper is ready to the publication, and therefore I recommend that it is published practically in its present form.

In saying that I would recommend to the authors to reconsider the title. Now it says "... surface plasmon..." which for many people associates with propagating plasmon. Strictly speaking the authors demonstrated coupling to the localized plasmon. In their answer to reviewer 1 (and now in supplementary materials) the authors convincingly argue that by adding some elements to the device, the propagating plasmon can be generated. This is OK, but strictly speaking the demonstration in the paper is done for the localised antenna mode. Actually I do not see the problem here because results are beautiful as they stand. The temptation to absolutely connect the findings to the propagating plasmons forces to operate with handwaving arguments, weakens the scientific rigor, and actually raises additional questions. Indeed: the photonic DOS for the propagating modes will be different and would not allow choosing the frequency, it actually might lead to the lower Purcell factor and thus to affect the efficiency of the conversion, and so on ...

Why not stating in the title explicitly that the work operates with localised antenna plasmon which corresponds to the reality?

Response to Reviewer #1

Initial statement: “The paper has been revised and the main claim partially changed. I recommend publication after a few mandatory changes.”

Reply: We thank the reviewer for kindly providing comments/suggestions on our manuscript. A point-by-point response to these comments/suggestions follows.

Comment 1: There is still a confusion in the paper already pointed out by referees 1 and 2 in the first round. The motivation put forward is plasmonic circuitry which requires surface plasmons polaritons (SPPs) whereas the EQE reported here only deals with localized surface plasmons (LSPs). The process of transforming a LSP into a SPP comes with a coupling cost. Using a nanowire supporting SPPs is not shown according to the supplementary because of low signal. Thus, this is not a detail. It is necessary to point out that the EQE reported here deals with LSP so that SP must be replaced everywhere in the text by LSP when it refers to the nano rods.

Reply: We thank the reviewer for the suggestion. We have clearly pointed out that we focus on the LSP in this work in the revised manuscript:

- 1) We have changed the title to “Highly-efficient electrically-driven **localized** surface plasmon source enabled by resonant inelastic electron tunneling”. We have also updated the introduction part in the manuscript accordingly.
- 2) We have given a specific explanation on the use of the LSP in the manuscript: “*In this Letter, we demonstrate the first on-chip electrically-driven resonant inelastic electron tunneling (RIET)-based SP source at the visible/NIR frequencies in metallic quantum well (MQW)-based tunnel junctions. Without loss of generality, we focus on the localized surface plasmon (LSP) source in the current RIET demonstration; the same RIET mechanism is readily applicable for non-LSP sources such as the surface plasmon polariton (SPP) source; moreover, with proper nanoantenna design¹⁵, one might also achieve high-performance RIET-based photon sources.*”
- 3) We have briefly outlined the future work about the selection of optical modes: “*Future optimization includes selecting less lossy plasmonic materials, improving the barrier material quality, and reducing the fabrication related defects, which will*

further boost the EQE. Various application-oriented RIET-based LSP, SPP and photon sources could be realized by designing proper optical micro/nanostructures.”

Also, we have replaced “SP” with “LSP” when it refers to the nanorods in the revised manuscript, as the reviewer suggested.

Comment 2: A second necessary modification is to discuss the uncertainty on the EQE estimation. There are no bar errors in the discussion in the supplementary.

Reply: We thank the reviewer for the suggestion. We have added the additional discussion on the measurement uncertainty in the revised supplementary information section S5 accordingly.

Response to Reviewer #2

Statement: After studying the responses of the authors to my criticisms, I could see that the changes introduced into the revised version are minor: there is still the efficiency (EQE) claim of 30%, which is now found not in the abstract but in the main text and Fig. 3c, still supported only by simulations indicating a very low far-field radiation efficiency. As was emphasized in my review, there are no propagating surface plasmons being excited, only localized ones, but the authors still write on p. 9 (and implicitly assume elsewhere) that “electrically-driven SPs were generated, propagated, and scattered at the end of the AgNRs”. The prospects of integration with a plasmonic circuitry are still only prospects described in a very short and highly speculative Supplementary Section S7, which is based on the prospect of realizing an RIET source based on a single AgNR. The latter would indeed be a very important development, but its realization is still missing. Finally, the bandwidth measurements in any form are still missing. Overall, in my opinion, the level of the obtained results is not sufficient for this work to be recommended for the publication in Nature Communications.

Reply: We thank the reviewer for his/her comments, and our response to these comments is as follows.

- (1) The EQE mentioned in the manuscript has been updated with the additional clear explanation.
- (2) We have modified and clearly stated the “localized surface plasmon” in the revised manuscript to avoid the confusion. See Reply to Comment 1 of Reviewer #1 and Reply to Comment of Reviewer #3.
- (3) The far-field measurement combining with the simulation of far-field radiation efficiency has widely been used to determine the EQE of inelastic electron tunneling (IET)-based photon/plasmonic sources. For example, recent works on IET photonic sources: Nat. Photonics 9, 582 (2015), Nat. Nanotechnol. 10, 1058 (2015), Nat. Photonics 12, 485 (2018), Nat. Commun. 10, 292 (2019); recent works on IET plasmonic sources: Nat. Photonics 10, 274 (2016), Nat. Photonics 11, 623-627 (2017), Nat. Commun. 10, 4949 (2019).
- (4) The operation speed and bandwidth of the inelastic electron tunneling (IET)-based photonic/plasmonic source are known to be the one of the intriguing properties in this

field. People in this community commonly believe the major experimental limitation of the IET-based surface plasmon devices would be from the driving circuit's RC response. As our current work focuses on the first demonstration of RIET in a metallic quantum wells system, the high-speed measurement is not essential.

Response to Reviewer #3

Initial statement: As I stated previously, the work of H. Qian et al reports on a very interesting experimental observation of the plasmon excitation in the tunneling metal-dielectric-metal (quantum well)-dielectric-metal device, where the elastic tunneling is suppressed and only the inelastic tunneling is possible. In an ideal system without losses each tunneling event can lead to the electromagnetic excitation, and, through the careful design of the electromagnetic density of states, to the plasmon excitation of the nanoantenna.

While the resonant structures alike are known in semiconductor physics (obviously at much lower electromagnetic radiation frequencies), present application for metal plasmonics is a primer. Already at the very beginning, I strongly supported publication of this work for the beauty of the physics and for the thoughtful design of the structure. Now, when the authors addressed comments of the referees twice, I think that the paper is ready to the publication, and therefore I recommend that it is published practically in its present form.

Reply: We appreciate the reviewer's recognition of the importance of this work and his/her strong recommendation for publication.

Comment: In saying that I would recommend to the authors to reconsider the title. Now it says "... surface plasmon..." which for many people associates with propagating plasmon. Strictly speaking the authors demonstrated coupling to the localized plasmon. In their answer to reviewer 1 (and now in supplementary materials) the authors convincingly argue that by adding some elements to the device, the propagating plasmon can be generated. This is OK, but strictly speaking the demonstration in the paper is done for the localised antenna mode. Actually I do not see the problem here because results are beautiful as they stand. The temptation to absolutely connect the findings to the propagating plasmons forces to operate with handwaving arguments, weakens the scientific rigor, and actually raises additional questions. Indeed: the photonic DOS for the propagating modes will be different and would not allow choosing the frequency, it actually might lead to the lower Purcell factor and thus to affect the efficiency of the conversion, and so on ...

Why not stating in the title explicitly that the work operates with localized antenna plasmon which corresponds to the reality?

Reply: We are grateful for the reviewer's comment and suggestion. We have modified and clearly stated the "localized surface plasmon" in the revised manuscript to avoid the confusion as the reviewer commented. And, we have modified the title as "*Highly-efficient electrically-driven localized surface plasmon source enabled by resonant inelastic electron tunneling*", according to the reviewer's suggestion.

REVIEWERS' COMMENTS

Reviewer #1 (Remarks to the Author):

I recommend to publish the paper in its present form.